# Boosting Adversarial Transferability with Shallow-Feature Attack on SAR Images

Gengyou Lin [1], Zhisong Pan [1,*], Xingyu Zhou [2], Yexin Duan [3], Wei Bai [1], Dazhi Zhan [1], Leqian Zhu [1], Gaoqiang Zhao [1] and Tao Li [1]

1 Command and Control Engineering College, Army Engineering University of PLA, Nanjing 210007, China; lingengyou123@aeu.edu.cn (G.L.); baiwei@aeu.edu.cn (W.B.); zhangaga93@aeu.edu.cn (D.Z.); leqianzhu@aeu.edu.cn (L.Z.); mrzgq@aeu.edu.cn (G.Z.); bin2vec@aeu.edu.cn (T.L.)
2 Communication Engineering College, Army Engineering University of PLA, Nanjing 210007, China; universezhou@aeu.edu.cn
3 Zhenjiang Campus, Army Military Transportation University of PLA, Zhenjiang 212000, China; yexin_0713@aeu.edu.cn
* Correspondence: panzhisong@aeu.edu.cn

**Abstract:** Adversarial example generation on Synthetic Aperture Radar (SAR) images is an important research area that could have significant impacts on security and environmental monitoring. However, most current adversarial attack methods on SAR images are designed for white-box situations by end-to-end means, which are often difficult to achieve in real-world situations. This article proposes a novel black-box targeted attack method, called Shallow-Feature Attack (SFA). Specifically, SFA assumes that the shallow features of the model are more capable of reflecting spatial and semantic information such as target contours and textures in the image. The proposed SFA generates ghost data packages for input images and generates critical features by extracting gradients and feature maps at shallow layers of the model. The feature-level loss is then constructed using the critical features from both clean images and target images, which is combined with the end-to-end loss to form a hybrid loss function. By fitting the critical features of the input image at specific shallow layers of the neural network to the target critical features, our attack method generates more powerful and transferable adversarial examples. Experimental results show that the adversarial examples generated by the SFA attack method improved the success rate of single-model attack under a black-box scenario by an average of 3.73%, and 4.61% after combining them with ensemble-model attack without victim models.

**Keywords:** adversarial attack; deep neural network; black-box attack; targeted attack; feature-level attack

## 1. Introduction

A Synthetic Aperture Radar (SAR) is a sensor that uses electromagnetic waves for imaging, works in the microwave band, and has been developed for half a century. It has strong information-gathering capabilities and the advantage of working around the clock and in all weather conditions, making it an essential imaging device today. SAR Automatic Target Recognition [1–3] (SAR-ATR) is a fundamental problem and challenge in SAR image recognition. It is widely used in the fields of mapping, surveillance, and environmental monitoring, and has great practical significance in military and battlefield situations. Due to the high risk of national security applications, SAR-ATR technology needs to be not only accurate but also highly reliable and secure. Therefore, SAR-ATR has received great attention in the past few decades. However, similarly to optical images, SAR images are vulnerable to adversarial attacks.

Deep neural networks (DNNs) have achieved excellent performance in many remote sensing applications, such as object detection [4–9], image classification [10–14], and semantic segmentation [15–20]. However, DNNs have been shown by Szegedy [21] to be

vulnerable to adversarial examples. Attackers can cause machine learning algorithms to classify images incorrectly by adding perturbations to images, which can have serious consequences in applications such as object detection and recognition. By using carefully crafted adversarial attack methods such as FGSM [22] and PGD [23], attackers can generate intentionally designed perturbations in images. These perturbations are usually imperceptible to the human visual system but can fool DNNs, resulting in an incorrect classification with high confidence. Undoubtedly, this phenomenon poses a threat to the security of deep learning-based image recognition systems in practical applications.

Deploying adversarial examples in SAR-ATR is a relatively new research field. For example, attackers can manipulate SAR images containing military facilities to hide or distort critical information, or manipulate images of environmental disasters to mislead rescue efforts. Existing research on adversarial attacks on SAR images mainly focuses on the white-box attack setting. However, it is usually impossible to obtain detailed information about the victim model in the real world. Therefore, conducting a black-box attack is more feasible, although it is more challenging. A possible solution is to conduct a white-box adversarial attack using a substitute model or construct an ensemble model, and then feed the generated adversarial examples back to the victim model. However, when there are differences in the model architecture between the training model and the victim model, the transferability of the adversarial examples is likely to be limited.

Our work is inspired by the works of Xu et al. [24], Yosinski [25], and Wang et al. [18], which suggest that different models may produce similar feature representations in the shallow layers of the neural network. Xu et al.'s work [24] generated universal adversarial perturbations based on black-box untargeted attacks, Wang et al. [26] aimed to generate more transferable untargeted adversarial attack examples, while targeted attacks are often more challenging, realistic and relevant. For example, in military scenarios, an attacker may want to disrupt a specific radar detection device rather than causing misclassification. Therefore, targeted adversarial attack research is necessary to enhance the robustness of deep learning models and defend against targeted attacks that may exist in real-world environments. Inspired by Xu et al.'s work [24], which found common vulnerabilities among different networks by attacking the features in the shallow layer of a given surrogate model, we believe that, compared to more abstract deep features, shallow features often retain more detailed spatial and semantic information of the image (such as object contours and textures) and share similar features, even in networks of different architectures. The similarity of shallow features, to some extent, promotes similar model classification decisions. Based on the above analysis, this article aims to conduct a black-box targeted attack on SAR images and generate adversarial examples with high transferability. In the absence of knowledge about the victim model, attacks can be carried out on different deep neural networks with high success rates. To better test the defensive performance and robustness of the SAR-ATR model, this paper proposes the Shallow-Feature Attack (SFA) method. Specifically, for a deep neural network, SFA first constructs ghost data packages using a 0–1 random mask in traditional image processing for the target and the clean image, then chooses the first pooling layer of different networks as the feature layer in our work to extract the aggregated gradients and feature maps of the input image at that layer, respectively, to construct the critical features. From this, SFA has completed the construction of the feature-level loss function. Next, we construct the end-to-end cross-entropy loss function for attack by minimizing the Euclidean distance between the logits of the input image and the target classification while maximizing the distance between the logits of the input image and the original classification. Finally, we define the hybrid loss function of SFA attack by combining the feature-level loss and the end-to-end loss. Through our designed SFA method, the generation process of adversarial examples will pay more attention to fitting the data in the specified feature space, while reducing the risk of falling into the model-specified local optimum, thus effectively improving the attack performance as well as transferability of adversarial examples in black-box scenarios.

In summary, the main contributions of this paper are as follows:

1. This article proposes the SFA attack, which utilizes ghost data packages to extract and fit critical target features that influence model decisions, to enhance the attack effectiveness of adversarial examples under black-box scenarios.
2. Unlike existing feature-based adversarial attacks, we focus on more realistic and challenging black-box targeted scenarios and consider the pooling layer that better reflects spatial and semantic information in the image (such as object contours and textures) as the target of feature-level attacks in different networks.
3. Extensive single-model and ensemble-model attacks on different classification models show that the adversarial examples generated by our proposed SFA method have stronger attack performance and better transferability.

The remaining structure of the paper is as follows: Section 2 mainly introduces related work on adversarial examples. Section 3 provides a detailed explanation of our proposed method. Section 4 presents experiments comparing the adversarial examples generated by our method with other attack methods under single-model and ensemble-model attacks. The conclusions and future work are presented in Section 5.

## 2. Related Works

### 2.1. Adversarial Attack in Deep Learning

For an initial clean image $x$ with a ground-truth label of $y^{true}$ and a pre-trained model classifier of $f(\cdot)$, the initial image should be correctly classified, i.e., $f(x) = y^{true}$. When the attacker adds a carefully designed noise $\delta$ to the initial clean image, a new image $x^{adv}$ is generated, i.e., $x^{adv} = x + \delta$, resulting in a new classification result for the model classifier, then $f\left(x^{adv}\right) = y^{adv}$. Here, $y^{adv}$ is the output classification of $x^{adv}$ in the model classifier $f(\cdot)$, in order for the adversarial example not to affect human recognition, and the attacker usually uses $L_p$ norm to constrain $x^{adv}$ within the neighborhood of $x$, i.e., the allowed adversarial perturbation should be less than a threshold value $\varepsilon$ that $\left\|x^{adv} - x\right\|_p \leq \varepsilon$, where $p = 0, 1, 2 \ or \ \infty$. Depending on the different attack objectives, two attack methods can be introduced: if $y^{adv} \neq y^{true}$, then $x^{adv}$ is called a untargeted adversarial example; if the attacker wishes the model to classify $x^{adv}$ as a specified target $y^{tar}$, and if $y^{adv} = y^{true}$, then $x^{adv}$ is called a targeted adversarial example.

Moreover, according to different threat models, adversarial attacks can also be classified as white-box attack, gray-box attack [27], and black-box attack. White-box attacks refer to attackers who have complete access to the internal structure, parameters, and even the training data distribution of the model. Attackers can probe vulnerable feature spaces of the model with available information and can achieve high success rates. This is a very strong adversarial attack. Gray-box attacks refer to attackers who only know everything about the model except for the model parameters, such as structure, hyperparameters, training data, etc., which is equivalent to having complete information about an untrained target model. Black-box attacks refer to attackers who are unfamiliar with the model and have no access to the internal information of the model, and they can only get information about the model input and output.

### 2.2. White-Box Attack Method

Gradient-based adversarial example generation methods are a commonly used white-box attack method, which mainly involves using the model loss function to calculate the gradient of the input image, and updating the image through back-propagation to generate adversarial examples. This section will focus on several gradient-based attack methods.

Goodfellow et al. [22] believed that linear features in high-dimensional space were sufficient to generate adversarial examples, and proposed a single-step gradient-based method called fast gradient sign method (FGSM) that only calculates the gradient for one

time, and amplifies the adversarial noise linearly in the direction of the obtained gradient using the sign function. FGSM can be expressed as:

$$x^{adv} = x + \varepsilon \cdot sign(\nabla_x J(x, y)), \tag{1}$$

where $sign(\cdot)$ is the sign function, $J(\cdot)$ is the loss function, $\nabla_x$ is the partial derivative of $x$ and the obtained adversarial example perturbation satisfies the norm distance constraint. The prominent advantage of FGSM is that it has a low computational cost, but its white-box attack success rate is not high, and its black-box transfer attack success rate is even lower.

Kurakin et al. [28] transformed FGSM to an iterative version named iterative fast gradient sign method (I-FGSM) with a smaller step size for iterative attack. It can be expressed as:

$$x_{t+1}^{adv} = Clip_{x_t^{adv}} \left\{ x_t^{adv} + \alpha \cdot sign \cdot \nabla_{x_t^{adv}} J\left( x_t^{adv}, y \right) \right\}, \tag{2}$$

where $t$ is the number of iterations, $x_t^{adv}$ indicates the adversarial example generated in the $t$-th iteration and $\alpha$ is the step size. In the white-box attack scenario, I-FGSM performs better than FGSM, but its transfer attack success rate is lower.

Momentum iterative fast gradient sign method (MI-FGSM) [29] combines the momentum term with I-FGSM in order to stabilize the update direction, overcome the drawback of falling into local maximum values, alleviate the overfitting problem, and significantly improve the success rate of adversarial examples. MI-FGSM can be expressed as:

$$g_{t+1} = \mu \cdot g_t + \frac{\nabla_{x_t^{adv}} J\left( x_t^{adv}, y \right)}{\left| \nabla_{x_t^{adv}} J\left( x_t^{adv}, y \right) \right|_1}, \tag{3}$$

$$x_{t+1}^{adv} = Clip_{x_t^{adv}} \left\{ x_t^{adv} + \alpha \cdot sign(g_{t+1}) \right\}, \tag{4}$$

where $g_t$ is the accumulated gradient vector of the loss with a momentum factor $\mu$. Subsequent work replaced MI-FGSM with the Nesterov optimization algorithm [30] to accelerate the gradient. MI can also be combined with DI [31] and TI [32] to form more powerful attacks.

### 2.3. Black-Box Attack Methods

Black-box attacks are usually divided into transfer-based attacks and query-based attacks. The transfer-based attack idea stems from the transferability of adversarial examples [21]. Adversarial examples generated by attacking a specific white-box model can also be used to attack another black-box model that performs the same task. Transfer-based attack does not rely on internal information of the target model. Instead, they train alternate models that have similar decision boundaries to the target model using the same or similar training data [33], and then generate adversarial examples using white-box attacks on the alternate models. Finally, they use the cross-model transferability of adversarial examples to attack the target model. Zhou et al. and Huang et al. found that attacks on the intermediate layers of the model are more powerful than attacks on the predicted logits [34,35]. Chen et al. [36] proposed conducting adversarial attacks on the attention maps of input images, which can achieve better results. Peng et al. [37] proposed the SVA attack, which consists of two major modules, an iterative gradient-based perturbation generator and a target region extractor, that can generate more transferable adversarial examples. Deepfool [38] is a transfer-based adversarial attack algorithm that aims to generate minimal perturbations to an input sample in order to mislead a neural network model. The ensemble-based method is a typical transfer-based attack method. The ensemble-based method [39] applies the idea of ensemble to adversarial example generation. It combines the logits (i.e., the output of the fully connected layer) of multiple pre-trained models, i.e., it combines the predicted values of multiple models, so that the generated adversarial examples can better

break the decision boundary of the target model and improve the attack success rate. The expression of the ensemble logits is:

$$f(x^{adv}) = \sum_{k=1}^{K} \omega_k \cdot f_k(x^{adv}), \tag{5}$$

where $f_k(x^{adv})$ is the logit of the $k$th model for the adversarial example $x^{adv}$ during the attack process, and $\omega_k$ is the model weight coefficient, $\omega_k \geq 0$ and $\sum_{k=1}^{K} \omega_k = 1$. Ensemble-based attack can be combined with methods such as FGSM, I-FGSM, MI-, NI-, etc.

Query-based attacks first add initial perturbations to images, and then continuously adjust the pixel values of adversarial examples by directly querying the output of the target model until convergence or the maximum iteration number is reached. Due to the need for a large number of queries, the computational cost is high, which is not conducive to generating a large number of adversarial examples. Query-based attacks pay more attention to controlling the perturbation of a single adversarial example and can further be divided into score-based attacks [40,41] and decision-based attacks [42]. The difference between the two lies in the fact that for the former, the target model sequentially outputs multiple predicted class labels and corresponding scores, and the attack can use the predicted scores, while for the latter, the target model only outputs the final decision class label and does not provide predicted scores, such as only the top-1 classification result without probabilities.

### 2.4. Feature-Level ATTACK methods

Zhou et al. [35] first proved that maximizing the feature distance between input images and their adversarial examples in the intermediate layer can enhance the transferability of images. Huang et al. [36] fine-tuned existing adversarial examples and added perturbations to the specified layer of the source model to achieve higher transferability. Ganeshan et al. [43] proposed a principled paradigm to disrupt feature representations for higher transferability. Xu et al. [24] analyzed the universal adversarial examples in remote sensing images for the first time, generating universal adversarial perturbations based on black-box untargeted attacks named Mixup-Attack. Our proposed method belongs to the category of attacking the internal features of the model, as well. This feature of our method lies in separating the image features in the model into critical features which play an important role in the model decision and model-specified features, which enhances the dominant features of the model decision in different models and suppresses the specific features that are prone to causing the model to fall into unique local optima.

### 2.5. Adversarial Attack on SAR Images

Existing studies on adversarial example generation methods are mostly limited to optical images, with only a few studies attempting to generate adversarial examples on SAR images [33,44–52]. In these studies, DNN-based SAR-ATR has been shown to be very vulnerable to small perturbations, and adversarial attacks can effectively manipulate SAR images. Meng et al. [44] proposed a local SAR-ATR adversarial deception algorithm that utilizes the Gabor feature-based texture segmentation (GFTS) method to extract the mask of the target area in SAR images. The mask is then introduced as a parameter into the loss function of the perturbation generator, thereby aggregating the adversarial perturbations of SAR samples into the target area. Zhang et al. [45] proposed a novel SAR-ATR adversarial deception algorithm that fully considers the characteristics of SAR data. The excellent performance of the algorithm was evaluated using four metrics: fooling rate, confidence score, structural similarity index (SSIM), and number of disturbed pixels (NDP). The two articles mentioned above are undoubtedly excellent achievements, playing a crucial role in the research of SAR image adversarial attacks. Our article mainly focuses on black-box targeted attacks. The SFA method proposed in our article pays more attention to the design of the loss function at the feature level compared to the above two methods. Czaja et al. [46] were the first to reveal the existence of adversarial examples in remote sensing image classification tasks. Their experiments showed that applying small adver-

sarial perturbations to remote sensing images can fool deep learning models into making incorrect classifications. Based on optical and SAR images, Chen et al. [47] conducted empirical research on adversarial examples in scene classification. They observed an interesting phenomenon that the adversarial examples produced on SAR images often have higher transferability across different models than those on optical images, indicating that SAR-ATR models are more vulnerable to adversarial attack. Xu et al. [48] further discovered the existence of adversarial examples in hyperspectral imaging. Their experiments showed that adversarial attacks can successfully change the spectral reflection characteristics of adversarial hyperspectral examples. Li et al. [49] comprehensively evaluated the adversarial vulnerabilities in SAR scene classification and target recognition tasks using existing optical attack methods, such as fast gradient sign method (FGSM), Carlini and Wagner (C&W) [53], and Deepfool [38], demonstrating that the prediction class of SAR adversarial images is highly concentrated. Du et al. [54] designed an accelerated C&W method to balance time consumption and attack capability. Wang et al. [26] proposed a feature aggregated gradient based on the feature selection method, which selected the features that maximize the attack effect.

Due to the relatively limited existing work on generating adversarial examples for SAR images and the lack of relevant work on targeted black-box scenarios, this paper intends to use the four methods mentioned above, which have related research on SAR images, as baseline methods. Table 1 classifies the four attack methods mentioned above.

**Table 1.** Mainstream adversarial attack algorithms. These attack methods mainly target digital scenes and conduct attacks in a human-imperceptible form.

| Attack Method | Threat Model | Target | Algorithm |
|---|---|---|---|
| FGSM [22] | white-box | untargeted | gradient-based |
| I-FGSM [28] | white-box | untargeted | gradient-based |
| MI-FGSM [29] | white-box | untargeted | gradient-based |
| Ens-Attack [39] | black-box | / | gradient-based |
| SVA [37] | black-box | untargeted | transfer-based |
| Mixup-Attack [24] | black-box | untargeted | feature-based |
| Deepfool [38] | black-box | / | transfer-based |

## 3. Methods

DNNs tend to extract semantic features which have discriminative ability for target awareness, effectively helping the model to improve classification accuracy. In theory, if all target-aware features that can affect model decisions in the input image are similar to the target class, it is easier for us to generate adversarial examples of the specified class. However, different networks may have different data features in the solution space for the same data domain, indicating the existence of specific feature representations of the model. However, existing adversarial attacks often only consider making the model misjudge by modifying gradients at the decision layer of the model, attacking images end-to-end indiscriminately. Such attacks may not always be successful, when facing complex black-box scenarios, their attack effects are often unsatisfactory, and they are easily trapped in model-specified local optima, greatly reducing the transferability of adversarial examples. Therefore, avoiding falling into such local optima is a key to improving the transferability of adversarial examples.

### 3.1. Overview

Based on this, this article proposes the SFA attack algorithm, and the flowchart of our attack method is shown in Figure 1.

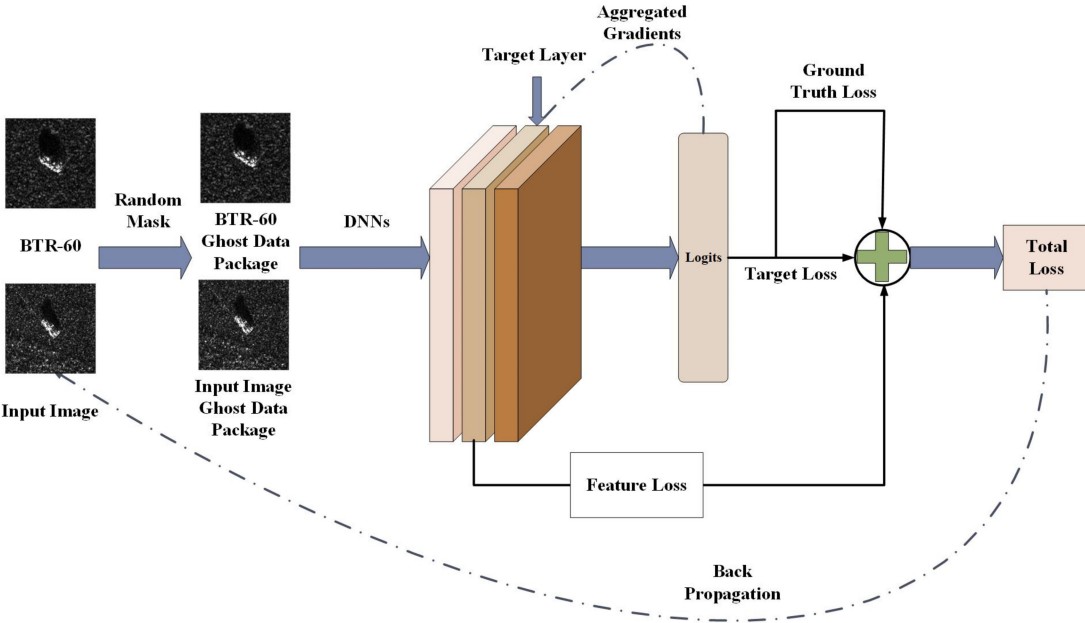

**Figure 1.** Flowchart of SFA algorithm.

The following is a brief introduction to our algorithm: first, the proposed SFA uses a 0-1 random mask to construct ghost data packages for the input image and target image (BTR_60 is set as our target label), and then perform gradient back-propagation in the model for the ground-truth class of the ghost data package, and intercept the back-propagated gradient at the first pooling layer to construct the aggregated gradient. SFA performs the dot product of the aggregated gradient with the feature map of the input data forward propagated at the first pooling layer to obtain the critical features of the input image for the model. In theory, if we can enable the critical features of the input image to fit the critical features of the target image, we can promote the model to classify the input image towards the target class, thus determining the loss function for feature-level attack. Additionally, SFA constructs an end-to-end loss function for the input image logits at the softmax layer of the model with respect to both the ground-truth class of the image and the target class we specify for end-to-end attack. The loss function proposed by our SFA method is obtained by adding the feature-level loss function and the end-to-end loss function.

In Sections 3.2 and 3.3, we will respectively discuss the two types of loss functions in detail.

### 3.2. Loss Function for Feature-Level Attacks

A good image recognition algorithm should be able to effectively extract target features and quickly locate new targets. Since different neural networks preserve more detailed semantic information (such as object contours and textures) in the shallow feature space and share similar features in different networks, these shallow feature spaces may contain more critical features that can affect the model's final decision. Considering that the importance of features is proportional to their contribution to the final decision, we hope to generate more powerful adversarial examples by attacking the features extracted by the shallow part of the neural network at the feature level. For targeted attack, we hope to enable the features extracted by clean examples in the shallow feature space of the deep neural network to fit the features extracted by examples towards the target class in the neural network.

However, as shown in Figure 2, we found that the raw feature maps extracted by the neural network in the shallow layers have noise in the visual sense, with noises and large gradients in non-object regions, which may be caused by the solution space of a specified model. In order to accurately extract the features that are truly critical factors

for the model's classification, rather than redundant and specific model features, we use the dot product of the feature map extracted for the specified layer and the gradient back-propagated to the specified layer to obtain the aggregated feature of that layer. Assuming that the input of the model is $x$, and the output of the first pooling layer is $a_{fp}(x)$, the output features propagated forward to the first pooling layer of the model are

$$f_{fp}(x) = a_{fp}(x) \tag{6}$$

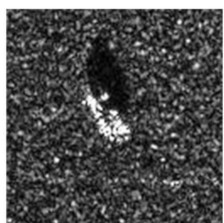 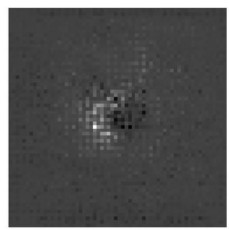 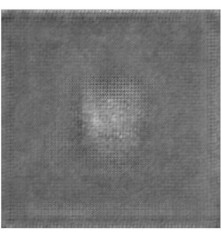 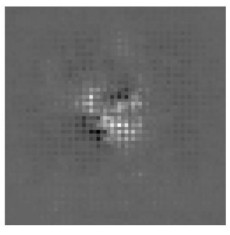 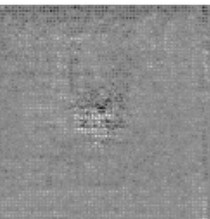

**Figure 2.** From left to right are the original image of BTR-60 and its feature maps extracted from the first pooling layer of AlexNet, DenseNet121, RegNet_x_400mf, and ResNet18.

Let $L$ be the loss function, $\omega_{ij}^{fp}$ be the weight from the $i$-th neuron of the first pooling layer to the $j$-th neuron of the next layer, $b_i^{fp}$ be the bias of the $i$-th neuron of the first pooling layer, $a_i^{fp}$ be the activation value of the $i$-th neuron of the first pooling layer, and $z_i^{fp}$ be the weighted output of the $i$-th neuron of the first pooling layer. According to the chain rule, we have:

$$\frac{\partial L}{\partial a_i^{fp}} = \sum_{j=1}^{n_{p+1}} \frac{\partial L}{\partial z_j^{p+1}} \frac{\partial z_j^{p+1}}{\partial a_i^{fp}}, \tag{7}$$

where $z_j^{p+1}$ is the weighted output of the $j$-th neuron in the next layer of the first pooling layer, $n_{p+1}$ is the number of neurons in the next layer of the first pooling layer. According to the definition of a neuron, we can obtain:

$$z_j^{p+1} = \sum_{i=1}^{fp} \omega_{ij}^{fp} a_i^{fp} + b_j^{fp} \tag{8}$$

Therefore, there are:

$$\frac{\partial z^{p+1}}{\partial a_i^{fp}} = \omega_{ij}^{fp} \tag{9}$$

Substituting (9) into (7) we have:

$$\frac{\partial L}{\partial a_i^{fp}} = \sum_{j=1}^{n_{p+1}} \frac{\partial L}{\partial z_j^{p+1}} \omega_{ij}^{fp} \tag{10}$$

Next, by deriving $\frac{\partial L}{\partial \omega_{ij}^{fp}}$ from $\frac{\partial L}{\partial z_j^{p+1}}$, we can obtain:

$$\frac{\partial L}{\partial \omega_{ij}^{fp}} = \frac{\partial L}{\partial z_j^{p+1}} \frac{\partial z_j^{p+1}}{\partial \omega_{ij}^{fp}} \tag{11}$$

Similarly, according to the definition of a neuron, we can obtain:

$$\frac{\partial z_j^{p+1}}{\partial \omega_{ij}^{fp}} = a_i^{fp} \tag{12}$$

Thus, it is possible to obtain:

$$\frac{\partial L}{\partial \omega_{ij}^{fp}} = \frac{\partial L}{\partial z_j^{p+1}} a_i^{fp} \tag{13}$$

In summary, we can obtain the gradient of the model back-propagation to the first pooling layer.

$$\frac{\partial L}{\partial a_i^{fp}} = \sum_{j=1}^{n_{p+1}} \frac{\partial L}{\partial z_j^{p+1}} \omega_{ij}^{fp}, \tag{14}$$

$$\frac{\partial L}{\partial \omega_{ij}^{fp}} = \frac{\partial L}{\partial z_j^{p+1}} a_i^{fp}, \tag{15}$$

where $\frac{\partial L}{\partial z_j^{p+1}}$ can be calculated using the gradient of the previous layer $\frac{\partial L}{\partial z_j^{fp}}$ and the derivative of the activation function. Then there is:

$$\nabla_{fp} x = \frac{\partial L}{\partial a_i^{fp}}, \tag{16}$$

where $\nabla_{fp} x$ denotes the gradient of the model back-propagated to the first pooling layer.

As shown in Figure 3, in this paper, SFA uses a random 0–1 mask with a probability of $p$ to perform image enhancement on the input image to form a ghost data package. Assuming that the ghost data package generated for each input image contains $N$ images, the aggregated gradient of the $N$ images can be represented by the following equation:

$$\overline{\nabla_{fp} x} = \frac{1}{N} \sum_{n=1}^{N} \nabla_{fp} x \odot M_p^n, M_p^n \sim Bernoulli(1-p), \tag{17}$$

where $M_p^n$ denotes the random mask operation performed on the $n$-th image in the ghost data package, and $\odot$ represents the vector dot product.

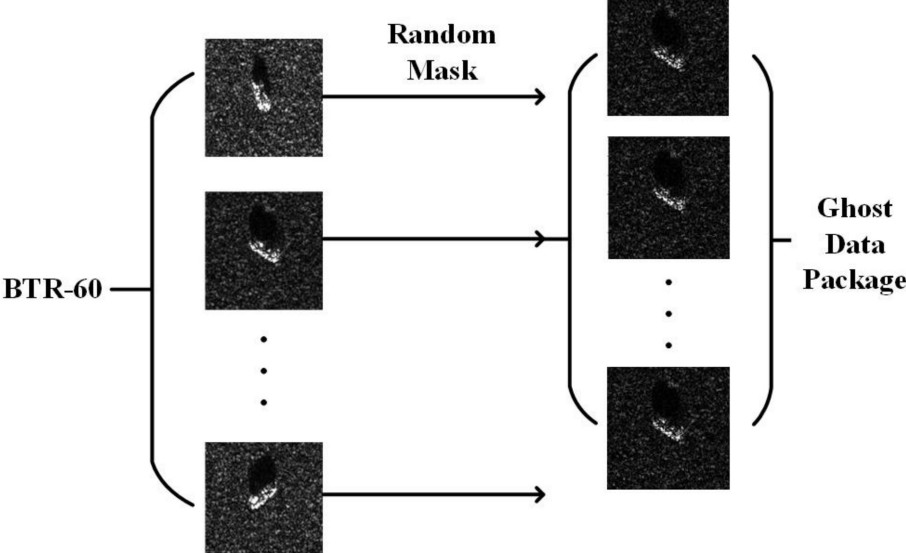

**Figure 3.** The process of generating ghost data packages through 0–1 random mask of input data.

To improve numerical stability and facilitate model training, we apply an L2 norm constraint to the aggregated gradients calculated in the SFA attack. By taking the dot product with the feature maps output by the first pooling layer, we calculate the critical features that have the strongest influence on the model decision for the target class in the

current neural network. Let $F$ represent the critical feature of the input image by the model at the first pooling layer, then we have:

$$F = f_{fp}(x) \cdot \left| \left| \overline{\nabla_{fp} x} \right| \right|_2 \tag{18}$$

Thus, we have implemented an approximate transformation of the image that distorts the details of the image, blurs the unique features of the model, but preserves the spatial structure and general texture of the image. Since the critical features of the semantic target are robust to this type of transformation, while the specific features of the model may be easily affected, those robust and transferable critical features will be highlighted after aggregation, while others will be weakened.

As shown in Figure 4, to enable the model to successfully misclassify the adversarial examples towards the target label by our method, SFA constructs ghost data packages for the input clean image and the target image, and computes their critical features in the classification model. We ensure that the critical features extracted from the clean image fit the critical features extracted from the target label in the same network. Therefore, we construct the first feature-level loss function: the KL loss. Let $L_{KL}$ represent the KL loss, $F_{tar}$ represent the critical features of the target class, $F_{adv}$ represent the critical features of the input data, and $X$ represent the set of points on the image. By normalizing the two features and converting them into approximate probability distributions, we can obtain the following expression:

$$L_{KL} = \sum_{x \in X} F_{adv}(x) \cdot log \frac{F_{adv}(x)}{F_{tar}(x)} \tag{19}$$

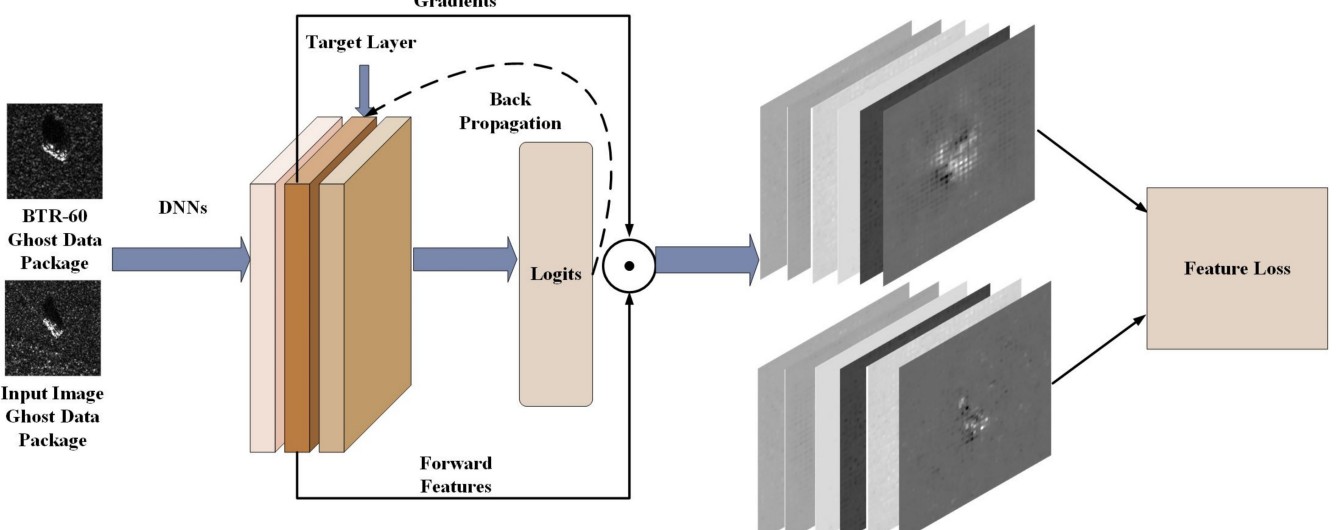

**Figure 4.** Critical feature loss process.

### 3.3. Loss Function for End-to-End Level Attacks

For a targeted attack, we want the output logits of the adversarial example to achieve the maximum value for the targeted class in the deep neural network. Let $f_\theta(x)$ denote the output of the input image $x$ in the model $f$, where $\theta$ represents the model parameter, $y^{true}$ is the ground-truth label of $x$, and $y^{tar}$ denotes the target class we want to classify. Therefore, we need to minimize the Euclidean distance between the output logits and the target class, which can be expressed as follows:

$$L_{tar} = \min \left\{ \left| \left| f_\theta(x) - y^{tar} \right| \right|_2 \right\} \tag{20}$$

At the same time, we hope to promote data crossing the decision boundary by making the output logits far from the ground-truth label of the image. That is, we maximize the Euclidean distance between the output logits and the ground-truth label, which can be expressed as follows:

$$L_{true} = \max\{||f_\theta(x) - y^{true}||_2\} \tag{21}$$

From this, we construct the loss function for the end-to-end level attack as follows:

$$L(x, y^{true}, y^{tar}, \theta) = \max\{||f_\theta(x) - y^{true}||_2 - ||f_\theta(x) - y^{tar}||_2\}, \tag{22}$$

$$L_{ce} = L(x, y^{true}, y^{tar}, \theta) \tag{23}$$

where the Euclidean distances of the output logits to both the ground-truth class and the target class are constrained by the L2 norm.

### 3.4. Total Loss Function

From Sections 3.2 and 3.3, our total loss function is designed as

$$L_{total} = L_{KL} + L_{ce} \tag{24}$$

It should be emphasized that our proposed SFA attack method defaults to a single-step iterative attack. However, similar to iterative methods such as I-FGSM and MI-FGSM, SFA can perform multi-step iterations or add momentum terms to form a more powerful attack. Furthermore, similar to ensemble model attacks, SFA can also be trained on multiple models to further improve the transferability of adversarial examples. Compared with the default SFA single-step iterative attack, after increasing the number of iterations, our SFA method can more accurately and in real-time capture critical features to achieve better attack effects. Below is the pseudocode for the Ens-I-SFA algorithm of multi-step iterative SFA under ensemble model (Algorithm 1).

---

**Algorithm 1**: Ens-I-SFA

| | |
|---|---|
| **Input:** | -An Ensemble Model $f_\theta$, |
| | -A clean image $x$ and its ground-truth label $y^{true}$, |
| | -An image of target label $y^{tar}$, |
| | -Step size $\alpha$, |
| | -Random mask probability $p$, |
| | -Number of iteration $N$ |
| **Output:** | The adversarial image $x^{adv}$ |
| **1:** | for *n = 0 to N-1* do |
| **2:** | Obtain critical features |
| | $\overline{\nabla_{fp}x} = \frac{1}{N}\sum_{n=1}^{N}\nabla_{fp}x \odot M_p^n, M_p^n \sim Bernoulli(1-p)$ |
| | $F = f_{fp}(x)\cdot\left\|\overline{\nabla_{fp}x}\right\|_2$ |
| **3:** | Construct feature-level loss: |
| | $L_{KL} = \sum_{x\in X}F_{adv}(x)\cdot log\frac{F_{adv}(x)}{F_{tar}(x)}$ |
| **4:** | Construct end-to-end-level loss: |
| | $L_{ce} = \max\{||f_\theta(x) - y^{true}||_2 - ||f_\theta(x) - y^{tar}||_2\}$ |
| **5:** | Construct hybrid loss: |
| | $L_{total} = L_{KL} + L_{ce}$ |
| **6:** | Update $x$ by iterative fast gradient sign method: |
| | $x_{n+1}^{adv} = Clip_{x_n^{adv}}\left\{x_n^{adv} + \alpha\cdot sign\cdot\nabla_{x_n^{adv}}L_{total}\right\}$ |
| **7:** | end for |
| **8:** | return $x^{adv}$ |

---



## 4. Experiments and Results

### 4.1. Experimental Preparation

We conducted model training and adversarial example generation on the MSTAR dataset, which is a dataset for Synthetic Aperture Radar Automatic Target Recognition (SAR-ATR) research launched by the Defense Advanced Research Projects Agency (DARPA) of the United States in 1998. The MSTAR dataset contains 10 different types of targets, namely: a 2S1 self-propelled howitzer, BMP2 armored infantry fighting vehicle, BRDM_2 reconnaissance patrol car, BTR_60 light transport vehicle, BTR_70 light transport vehicle, D7 bulldozer, T62 main battle tank, T72 main battle tank, ZIL131 truck, and a ZSU_23_4 self-propelled anti-aircraft gun, covering a variety of shapes, sizes, poses, and materials. SAR images in the MSTAR dataset were obtained under different weather and lighting conditions, which puts higher requirements on the robustness and generalization ability of target recognition algorithms. The MSTAR dataset contains thousands of SAR images, most of which are used for training and a small percentage of which are for testing. Each image contains a target and corresponding true annotation information, including the category, position, size, and pose of the target, etc. The MSTAR dataset has become a classic dataset in the field of target recognition and is widely used for algorithm evaluation and comparison. In the data preprocessing stage, we resized all images to 256*256 and normalized them.

For model selection, we used twelve typical SAR-ATR models. As shown in Table 2, all models achieved recognition accuracy of over 96% on the MSTAR dataset.

**Table 2.** Models used in our experiment with their test accuracy on MSTAR dataset.

| Model | Accuracy |
|:---:|:---:|
| AlexNet [55] | 96.3 |
| ResNet18 [56] | 96.8 |
| ResNet50 [56] | 96.4 |
| ResNet101 [56] | 97.5 |
| DenseNet121 [57] | 97.8 |
| DenseNet169 [57] | 97.9 |
| DenseNet201 [57] | 97.9 |
| RegNetX_400MF [58] | 98.3 |
| VGG11 [59] | 98.1 |
| VGG16 [59] | 97.8 |
| VGG19 [59] | 98.0 |
| Inception-v3 [60] | 98.1 |

### 4.2. Experiment Setup

From the tactical and strategic value of the target, BTR-60 and BTR-70 armored vehicles are usually used for transportation and support operations. Compared with other target classes, their combat power and strategic value are relatively low. Therefore, in this paper, the target attack will cause the sensor to identify the target as BTR-60. In our experiment, we define a successful attack as the target classifier misclassifying the adversarial examples generated from the source neural network as BTR-60. For the number of ghost data packages, we refer to the work of Wang et al. [26] and set it to 30. We used gradient-based attack methods (FGSM, I-FGSM, MI-FGSM) and transfer-based attack methods such as SVA, as well as its variants, Deepfool and Ensemble-Attack as the baseline method for this experiment.

Considering the low resolution of the images in the MSTAR dataset and the possibility of less semantic information contained, we set the size of the perturbation ball $\varepsilon$ to 1 in our work. For attacks other than FGSM, SFA and SVA, the number of iterations $N$ is set to 5. The step size $\alpha$ is set to 1 for all attack methods, and for MI-FGSM, MI-SVA and MI-SFA, the momentum decay $\mu$ is set to 0.5. For Deepfool, overshoot was set to be 0.02. Our experiments were performed on a server with the Ubuntu 20.04 operating system, Intel Xeon CPU E5-2609, and eight NVIDIA GTX 1080Ti, using Pytorch.

### 4.3. Single-Model Attack Experiments

We conducted single-model attack experiments on four typical SAR-ATR models, namely AlexNet [55], ResNet18 [56], DenseNet121 [57], and RegNetX_400MF [58]. The baseline methods used were FGSM, I-FGSM, and MI-FGSM, while the comparative methods were SFA, I-SFA, and MI-SFA. We tested the models on eight SOTA SAR-ATR models, including ResNet50 [56], ResNet101 [56], DenseNet169 [57], DenseNet201 [57], VGG11 [59], VGG16 [59], VGG19 [59], and Inception-v3 [60]. The experimental results are shown in Tables 3–6. Each group of three rows in the table corresponds to the success rates of attacks by SFA or its variants and the corresponding baseline FGSM and SVA methods, with the best results highlighted in bold.

**Table 3.** Success rate on different DNNs by different attack methods trained on AlexNet.

| Surrogate: AlexNet | Target Model | | | | | | | | Total |
|---|---|---|---|---|---|---|---|---|---|
| Attack | Res-50 | Res-101 | VGG11 | VGG16 | VGG19 | Dense-169 | Dense-201 | Inc-v3 | |
| FGSM | 3.96 | 3.07 | 4.24 | 3.22 | 3.35 | 3.42 | 2.80 | 2.33 | 26.39 |
| SVA | 4.54 | 3.59 | 4.97 | 3.96 | **4.03** | 4.14 | 3.33 | **3.02** | 31.58 |
| SFA | **4.99** | **4.04** | **5.71** | **5.17** | **4.03** | **6.38** | **4.48** | 1.35 | **36.15** |
| I-FGSM | 4.06 | 3.48 | 4.44 | 3.90 | 3.01 | 3.09 | 3.48 | 2.68 | 28.14 |
| I-SVA | 5.21 | 4.75 | 5.02 | 5.22 | 4.60 | 3.95 | 4.04 | 3.51 | 36.30 |
| I- SFA | **9.77** | **7.79** | **10.98** | **7.40** | **9.36** | **6.11** | **7.47** | **4.69** | **63.57** |
| MI-FGSM | 5.12 | 3.54 | 4.51 | 3.89 | 2.94 | 4.02 | 3.54 | 3.06 | 30.62 |
| MI-SVA | 6.58 | 5.23 | 5.91 | 5.26 | 4.41 | 5.34 | 4.86 | **4.14** | 41.73 |
| MI-SFA | **11.12** | **5.44** | **7.85** | **5.37** | **6.53** | **5.70** | **5.08** | 4.00 | **51.09** |
| Deepfool | 1.60 | 1.27 | 1.76 | 1.40 | 1.39 | 1.46 | 1.18 | 1.07 | 11.13 |

**Table 4.** Success rate on different DNNs by different attack methods trained on ResNet18.

| Surrogate: ResNet18 | Target Model | | | | | | | | Total |
|---|---|---|---|---|---|---|---|---|---|
| Attack | Res-50 | Res-101 | VGG11 | VGG16 | VGG19 | Dense-169 | Dense-201 | Inc-v3 | |
| FGSM | 3.69 | 2.94 | 3.28 | 3.14 | 3.28 | 3.35 | 1.85 | 0.34 | 21.87 |
| SVA | 5.02 | 4.23 | 4.74 | 4.67 | 4.66 | **4.36** | **2.77** | 1.30 | 31.75 |
| SFA | **8.93** | **7.33** | **7.80** | **9.50** | **6.11** | 3.43 | 2.73 | **2.86** | **48.69** |
| I-FGSM | 3.95 | 3.14 | 3.42 | 3.21 | 3.21 | **3.14** | 1.93 | 1.03 | 23.25 |
| I-SVA | 5.61 | 4.31 | 4.65 | 4.29 | 4.68 | **4.02** | **2.86** | 2.14 | 32.56 |
| I- SFA | **10.62** | **9.85** | **8.95** | **9.27** | **5.58** | 2.40 | 2.31 | **6.09** | **55.07** |
| MI-FGSM | 4.83 | 4.48 | 4.58 | 4.60 | 4.24 | 4.51 | 2.04 | 1.76 | 31.04 |
| MI-SVA | 6.60 | 6.01 | 5.64 | 6.02 | 5.40 | **5.63** | 3.26 | 2.99 | 41.55 |
| MI-SFA | **7.71** | **7.22** | **10.63** | **9.80** | **7.99** | 5.61 | **6.15** | **6.55** | **61.66** |
| Deepfool | 1.77 | 1.49 | 1.67 | 1.65 | 1.65 | 1.54 | 0.98 | 0.46 | 11.21 |

**Table 5.** Success rate on different DNNs by different attack methods trained on DenseNet121.

| Surrogate: DenseNet121 | Target Model | | | | | | | | Total |
|---|---|---|---|---|---|---|---|---|---|
| Attack | Res-50 | Res-101 | VGG11 | VGG16 | VGG19 | Dense-169 | Dense-201 | Inc-v3 | |
| FGSM | 3.42 | 2.67 | 3.35 | 3.08 | 3.08 | 2.46 | 1.78 | 0.89 | 17.65 |
| SVA | **4.41** | **4.08** | 5.08 | 4.77 | **4.60** | 3.51 | 2.50 | 1.66 | 30.61 |
| SFA | 4.01 | 3.34 | **5.61** | **5.97** | 4.17 | **6.77** | **4.16** | **2.45** | **36.48** |
| I-FGSM | 1.16 | 0.55 | 3.28 | 3.14 | 3.08 | 2.49 | 2.03 | 1.23 | 16.96 |
| I-SVA | **2.44** | 2.15 | 4.68 | 4.41 | 4.37 | 3.42 | 2.86 | 2.36 | 26.69 |
| I- SFA | 1.12 | **4.44** | **8.52** | **8.04** | **7.51** | **6.37** | **4.09** | **5.41** | **45.50** |
| MI-FGSM | 1.40 | 0.57 | 3.45 | 3.42 | 2.99 | 2.68 | 2.10 | 1.03 | 17.64 |
| MI-SVA | 2.95 | 2.45 | 4.89 | 4.79 | 4.49 | 3.82 | **3.37** | 2.35 | 29.02 |
| MI-SFA | **5.47** | **4.02** | **5.33** | **7.25** | **5.51** | **4.35** | 2.05 | **3.61** | **37.59** |
| Deepfool | 1.56 | 1.44 | 1.79 | 1.68 | 1.62 | 1.24 | 0.88 | 0.59 | 9.18 |

**Table 6.** Success rate on different DNNs by different attack methods trained on RegNetX_400MF.

| Surrogate: RegNetX_400MF | Target Model | | | | | | | | Total |
|---|---|---|---|---|---|---|---|---|---|
| Attack | Res-50 | Res-101 | VGG11 | VGG16 | VGG19 | Dense-169 | Dense-201 | Inc-v3 | |
| FGSM | 3.35 | 3.08 | 3.14 | 3.08 | 3.21 | 3.21 | 3.01 | 1.85 | 24.04 |
| SVA | **4.40** | **4.20** | 4.83 | 4.45 | 4.76 | 4.33 | 3.66 | 2.55 | 33.18 |
| SFA | 3.58 | 2.94 | **8.33** | **6.82** | **3.63** | **5.29** | **4.54** | **5.96** | **41.09** |
| I-FGSM | 3.35 | 3.21 | 3.28 | 3.08 | 3.28 | 3.08 | 3.14 | 1.57 | 23.99 |
| I-SVA | 5.07 | 4.54 | 4.37 | 4.75 | 4.70 | **4.02** | **3.82** | 2.33 | 33.60 |
| I- SFA | **5.38** | **5.88** | **7.13** | **5.55** | **4.86** | 2.49 | 2.26 | **2.38** | **35.84** |
| MI-FGSM | 4.43 | 4.35 | 4.65 | 4.26 | 4.31 | 4.27 | 4.03 | 2.78 | 33.08 |
| MI-SVA | 5.93 | **5.78** | 5.71 | 5.64 | 5.34 | 5.24 | 5.16 | 4.21 | 43.01 |
| MI-SFA | **8.74** | 5.51 | **9.06** | **8.56** | **8.65** | **7.44** | **7.46** | **7.56** | **62.98** |
| Deepfool | 1.55 | 1.48 | 1.71 | 1.57 | 1.68 | 1.53 | 1.29 | 0.90 | 11.71 |

It can be seen from Tables 3–6 that in almost all black-box scenarios where the substitute models are different from the target models, the performance of the proposed SFA method is better than the corresponding baseline methods. Taking the results pertaining to AlexNet-ResNet50 (the former is the substitute model and the latter is the target model) as an example, the success rate of MI-FGSM is about 5.12%, the MI-SVA is about 6.58% while the success rate of MI-SFA is 11.12%, exceeding by 6% to MI-FGSM and 4.54% to MI-SVA. These results indicate that the SFA method proposed in this paper can generate targeted adversarial examples with strong transferability to different target models.

At the same time, it can be seen from the horizontal comparison of Tables 3–6 that the SFA algorithm generates adversarial examples attacking deep models relatively easily using shallow models as substitute models, and the performance of the generated adversarial examples also improves more. On the other hand, SFA has relatively more difficulty in generating adversarial examples attacking shallow models using deep models. We speculate that this is because deep models usually have more complex structures and more parameters, and are more sensitive to gradient changes caused by feature loss in high-dimensional linear space. Therefore, we believe that the reason for the success rate of the conventional baseline methods in adversarial attacks is that the adversarial examples are trapped in model-specific local optima. However, through the influence of feature-level loss on gradients in our SFA algorithm, the adversarial examples can effectively escape local optima and cross decision boundaries. Furthermore, it can be seen that although SFA and the baseline methods compared are all transfer-based attack methods, the attack effectiveness of the FGSM-based approach (SFA, SVA) is significantly higher than Deepfool.

In addition, from the vertical comparison within Tables 3–6, we can see that the SFA algorithm proposed by us has a greater improvement in multiple iterations of adversarial attacks than in single iterations (i.e., FGSM, SVA and SFA). We speculate that this is because in each round of iteration, we reconstruct ghost data packages and calculate the critical features of the current adversarial image based on the adversarial image generated in the last round of iteration. Therefore, we can accurately incorporate the feature-level constraints into the loss function when modifying the gradient of the adversarial example in each round of iteration, which ensures a higher likelihood of our adversarial examples escaping the specific local optimum of the model.

According to statistics, we have performed an operation of summing up and taking the average of the differences between the success rates of the adversarial examples generated by SFA, I-SFA, and MI-SFA trained on four substitute models (AlexNet, ResNet18, DenseNet121, and RegNetX_400MF) and the corresponding baseline methods (FGSM, I-FGSM, MI-FGSM, SVA, I-SVA, and MI-SVA) under different target models. Based on this, we have drawn the following conclusions: the attack success rate of targeted adversarial examples generated by the SFA algorithm under single-model attacks has increased by an

average of 3.73% compared to the corresponding baseline FGSM methods and an average of 1.71% compared to the SVA methods.

### 4.4. Multi-Model Attack Experiments

In this section, we constructed an ensemble model consisting of AlexNet, ResNet18, DenseNet121, and RegNetX_400MF, and conducted attacks on ResNet50, ResNet101, DenseNet169, DenseNet201, VGG11, VGG16, VGG19, and Inception-v3, respectively. The attack results are shown in Table 7.

**Table 7.** Success rate on different DNNs by different attack methods trained on ensemble model.

| Ensemble Model | Target Model | | | | | | | | Total |
|---|---|---|---|---|---|---|---|---|---|
| Attack | Res-50 | Res-101 | VGG11 | VGG16 | VGG19 | Dense-169 | Dense-201 | Inc-v3 | |
| FGSM | 3.42 | 3.08 | 3.14 | 3.08 | 3.21 | 3.28 | 3.01 | 1.35 | 23.57 |
| SVA | 4.91 | 3.95 | 5.10 | 4.60 | 5.01 | 3.91 | 3.70 | 2.45 | 33.63 |
| SFA | **5.18** | **4.60** | **6.25** | **6.01** | **5.86** | **5.23** | **4.31** | **3.26** | **40.70** |
| I-FGSM | 4.60 | 4.13 | 4.61 | 4.33 | 4.15 | 3.95 | 3.65 | 2.63 | 32.05 |
| I-SVA | 6.77 | 6.03 | 6.50 | 5.88 | 5.26 | 4.92 | 4.20 | 3.79 | 43.35 |
| I- SFA | **6.93** | **6.21** | **9.01** | **8.51** | **7.82** | **7.79** | **6.28** | **5.34** | **57.89** |
| MI-FGSM | 4.95 | 4.23 | 5.30 | 5.04 | 4.62 | 4.87 | 3.93 | 3.16 | 36.10 |
| MI-SVA | 6.60 | 6.08 | 7.23 | 6.88 | 5.83 | 5.39 | 4.47 | 4.06 | 46.54 |
| MI-SFA | **8.82** | **6.60** | **9.60** | **8.69** | **7.76** | **8.69** | **8.01** | **7.32** | **66.49** |
| Deepfool | 1.73 | 1.40 | 1.80 | 1.62 | 1.77 | 1.38 | 1.31 | 0.87 | 11.88 |

From Table 7, it can be seen that the SFA method based on the ensemble model generates adversarial examples and is compared with the corresponding baseline FGSM and SVA methods. Similarly, we calculated the attack success rates of SFA, I-SFA, and MI-SFA under different target models in the ensemble model, and compared them with the corresponding baseline methods. The average attack success rate of the ensemble model was found to be increased by 4.61% compared to the corresponding FGSM method, and by 1.26% compared to the corresponding SVA method. Compared with single-model attacks, the average success rate of adversarial examples generated by the SFA method based on the ensemble model is about 0.88% higher. We speculate that this is because the four models participating in the ensemble model have strengthened the important features that reflect the target information and promote model decisions at the feature level, while the features that reflect the characteristics of each model have naturally been weakened.

However, our ensemble attack did not show significant improvement compared to single-model attacks, and we think this is because ensemble model attacks, as a method of transfer learning, construct the loss function by obtaining the logits of different models and weighting them. Due to the nature of black-box scenarios, it is impossible to know the parameters of the target model. Therefore, existing work often seeks to improve the success rate of ensemble attacks by increasing the number or types of models in the ensemble. However, ensemble model attacks face an unavoidable problem: if the number of models in the ensemble is too large or the model pool does not include the victim model, the generated adversarial examples often have poor transferability to the victim model. In traditional ensemble model attacks, the models included in the ensemble model often include the victim model, while in our experiment, the four models participating in the ensemble model are not in the victim model pool.

### 4.5. Hyperparametric Research

In this section, we mainly discuss some hyperparameters on the performance of our method. First, we conducted a study on the impact of the proportion $p$ of the 0-1 random mask on the ghost data packages on the attack performance of the SFA method.

As a product of using 0-1 random masks to enhance images, the ghost data package should reflect more significant and critical features of the input images for the model.

If the value of $p$ is too large, such as $p = 1$, the image becomes blank and no longer contains any information. If the value of $p$ is too small, such as $p = 0$, the image does not change at all. Therefore, we hope to obtain empirical data on the value of $p$ as a support for future research. Based on the SFA basic attack method and ResNet18 as the substitute model, we used ResNet50, ResNet101, VGG11, VGG16, VGG19, DenseNet169, DenseNet201, and Inception-v3 as the victim models to evaluate the attack performance of different values of $p$. We believe that the value of $p$ cannot exceed 0.5. Therefore, we conducted comparative experiments with $p$ values of 0.1, 0.2, 0.3, 0.4, and 0.5, and the experimental results are shown in Figure 5.

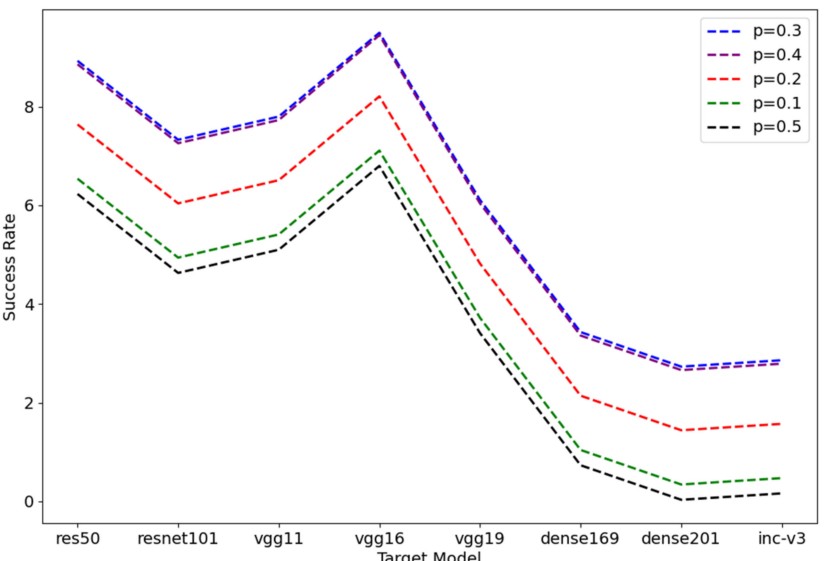

**Figure 5.** Attack success rates on ResNet18 with different 0–1 random mask rate $p$ against 8 test models using SFA.

From Figure 5, when $p = 0.1$, the ghost data package almost does not mask the pixels in the image, and the image enhancement effect is invalid in this case. When $p = 0.5$, the attack success rate is the lowest, and the attack effect is even worse than when $p = 0.1$. We speculate that a value of $p$ that is too large will destroy the important structural information of the image, making it impossible to extract critical features of the input image fully at the feature level, thus causing the input image to learn "false features". When $p = 0.3$ and $p = 0.4$, the attack success rate is basically the highest. Therefore, in our experiments in this paper, we set the probability of the 0–1 random mask to $p = 0.3$.

At the same time, the weights of feature-level loss and end-to-end loss in SAR adversarial training also need to be studied. From Tables 3–6, it can be seen that the attack success rate improvement of SFA and its variants is more significant when the target model is from the VGG series. However, when the target model is from the DenseNet series or Inception-v3, the attack success rate improvement of SFA and its variants is sometimes not significant enough, and even in some cases, the attack success rate of SFA or its variants is slightly lower than that of the corresponding baseline methods. It is speculated that this is because the weight coefficient of the feature-level loss is too large, which causes the model to overly focus on fitting shallow features when training adversarial examples. This behavior may cause the perturbation direction to deviate, making the data unable to cross the decision boundary.

Rewriting formula (24), we have

$$L_{total} = \alpha L_{KL} + (1 - \alpha)L_{ce} \tag{25}$$

where $\alpha$ is the weight coefficient of $L_{KL}$. Obviously, when $\alpha = 0$, the SFA algorithm degenerates into the FGSM algorithm. When $\alpha = 0.5$, it is consistent with formula (24).

Therefore, we discuss the values of $\alpha$ in the range of $[0, 0.5]$. We conducted experiments on DenseNet201 and Inception-v3, which are less effective in attack, using the SFA algorithm with a step size of 0.1. The experimental results are shown in the Table 8.

**Table 8.** Success rate on DenseNet201 and Inception-v3 by SFA with different $\alpha$ trained on AlexNet.

| SFA | DenseNet201 | Inception-v3 |
|:---:|:---:|:---:|
| $\alpha = 0$ | 2.8 | 2.33 |
| $\alpha = 0.1$ | 3.84 | 2.46 |
| $\alpha = 0.2$ | **4.85** | 2.50 |
| $\alpha = 0.3$ | 4.79 | **2.53** |
| $\alpha = 0.4$ | 4.73 | 1.97 |
| $\alpha = 0.5$ | 4.48 | 1.35 |

As shown in Table 8, when DenseNet is the target model, the success rate of SFA attack varies by no more than 0.37% when $\alpha$ is set between 0.2 and 0.5. However, even when the lowest value of $\alpha$ is used (i.e., $\alpha = 0.5$), the success rate of SFA attack is still about 1.64% higher than that of FGSM attack when $\alpha = 0$, indicating a significant improvement in attack effectiveness, which is consistent with the results for other models except Inception-v3. For Inception-v3, the selection of $\alpha$ values is more cautious, and only when $\alpha$ is set between 0.1 and 0.3, the SFA attack is more effective than the corresponding baseline method. Based on the characteristics of the model itself, we can only speculate that the Inception module may affect the effectiveness of the SFA algorithm, which will be explored in future work. Therefore, setting $\alpha$ to 0.2 is more appropriate in future research.

## 5. Discussion

From the experiments in Section 4, it is clear that there are still many aspects of our method that require further discussion. In this section, we will conduct a detailed analysis and discussion of each aspect.

### 5.1. Discussion on Target Setting for Targeted Attacks

As mentioned in Section 4.2, we determined the target category BTR_60 for the targeted attack in this experiment based on the tactical and strategic value of the target. However, in this section, we discuss whether our attacks can maintain similar effects when the target is set to the other nine categories. We sequentially set 2S1, BMP2, BRDM_2, BTR_70, D7, T62, T72, ZIL131, and ZSU_23_4 as the target category and conducted the targeted attack on VGG11, which performed the best in Section 4.4, using AlexNet as the source model. The experimental results are shown in the Table 9.

**Table 9.** Success rate on VGG11 by SFA with different target labels trained on AlexNet.

| Target Label | Success Rate |
|:---:|:---:|
| 2S1 | 9.55 |
| BMP2 | 10.19 |
| BRDM_2 | 11.48 |
| BTR_70 | 5.71 |
| D7 | 6.34 |
| T62 | 6.72 |
| T72 | 6.73 |
| ZIL131 | 6.99 |
| ZSU_23_4 | 10.83 |

According to Table 9, it can be seen that the success rate of our SFA method in attacking different categories of targets in the MSTAR dataset varies. Among them, when targeting 2S1, BMP2, BRDM 2, and ZSU 23 4, the attack success rate is almost twice as high as that of targeting other categories. The lowest attack success rate was achieved when targeting

BTR 60 and BTR 70. From Figure 6, it is not difficult to find that BTR 60 and BTR 70, as carrier trucks, have a rectangular shape that is too regular and lacks distinctive details specific to their category. On the other hand, other categories have more or less unique features. This may be the reason why BTR 60 and BTR 70 are relatively difficult to attack.

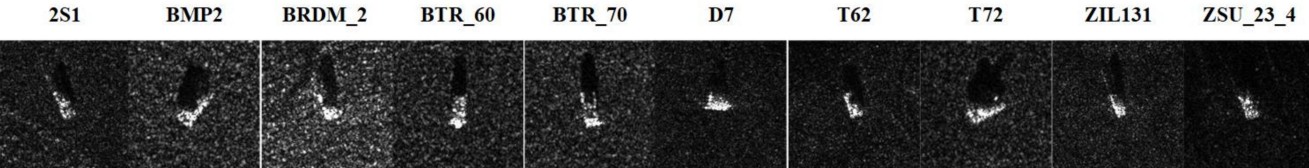

**Figure 6.** Different labels in MSTAR dataset.

### 5.2. Discussion on the Consumption of SFA with Its Variants

In this section, we want to explore the time consumption of SFA with its variants compared to the corresponding baseline models on different models. Corresponding to Tables 3–6, we provide the time consumption of the corresponding algorithms for generating adversarial examples on a single image on four substitute models.

As shown in Table 10, SFA or its variants have a time consumption that is roughly 30 times that of each corresponding baseline FGSM method and about 5 times that of each corresponding baseline SVA method. This is because the SFA method needs to create ghost data packets for the input data during the feature extraction phase and perform a complete gradient back-propagation for each image in the ghost packet. Therefore, the time consumed by the SFA method is proportional to the number of images in the ghost data packet compared to the time consumed by the baseline FGSM methods. In future work, we hope to explore ways to reduce the number of images in the ghost data packet while maintaining the same level of attack effectiveness, which would be an interesting and practical problem to solve.

**Table 10.** The time required in seconds for training adversarial examples on a single image using SFA with its variants, as well as the corresponding baseline methods, on four substitute models. The time consumption of SFA and its variants is highlighted in bold.

| Model<br>Attack | AlexNet | ResNet18 | Densenet121 | RegNetX_400MF |
|---|---|---|---|---|
| FGSM | 0.06 | 0.07 | 0.18 | 0.11 |
| SVA | 0.26 | 0.33 | 0.85 | 0.52 |
| SFA | **0.78** | **1.62** | **4.24** | **2.87** |
| I-FGSM | 0.13 | 0.18 | 0.62 | 0.62 |
| I-SVA | 0.22 | 1.04 | 3.58 | 1.04 |
| I-SFA | **3.75** | **8.89** | **22.23** | **14.84** |
| MI-FGSM | 0.12 | 0.17 | 0.62 | 0.55 |
| MI-SVA | 0.17 | 1.20 | 4.38 | 3.89 |
| MI-SFA | **3.29** | **6.48** | **20.99** | **13.81** |

### 5.3. Discussion of SFA on Conventional Machine Learning-Based Models

In the experiments in Section 4, we have fully demonstrated the powerful effectiveness of SFA and its variants in different deep neural networks, which can be considered as a cross-model transferability evaluation. However, we hope to further investigate our research by applying SFA and its variants to conventional machine learning-based models, and test the cross-technique transferability of SFA.

We trained a multi-class SVM model based on the MSTAR dataset and tested it using the adversarial examples generated by the SFA method and its variants trained on the source model AlexNet with the best performance in the single-model attack experiments in Section 4.3. As shown in Table 11, the attack effect of the adversarial examples on the multi-class SVM model is almost twice as effective as that on the target model VGG11.

**Table 11.** Success rate of SFA with its variants on VGG11 and SVM model.

| Model<br>Attack | VGG11 | SVM |
|---|---|---|
| SFA | 5.71 | 9.78 |
| I-SFA | 10.98 | 18.96 |
| MI-SFA | 7.85 | 13.03 |

Therefore, we believe that the SFA method not only has cross-model transferability but also has good cross-technique transferability. Meanwhile, it is not difficult to observe that non-differentiable models based on traditional machine learning such as SVM are more susceptible to adversarial examples compared with differentiable DNN models.

*5.4. Discussion of the Impact of Compression and Reconstruction*

As shown in Figure 7, due to the low resolution of SAR images and less semantic information in the background except for the target, we found through experiments that the pixel-level jitter of the image is more severe after the SFA attack on a SAR image compared to an optical image, especially in the target contour. When images are saved (such as saved as PNG format images), compression algorithms are often used to compress the images. Most compression algorithms use a DEFLATE algorithm, which tends to reduce the difference between large and small data in the images. This can weaken the adversarial nature of the adversarial images to some extent. Therefore, in this section, we will study the impact of compression and reconstruction methods in traditional image processing on the adversarial nature of SAR adversarial images.

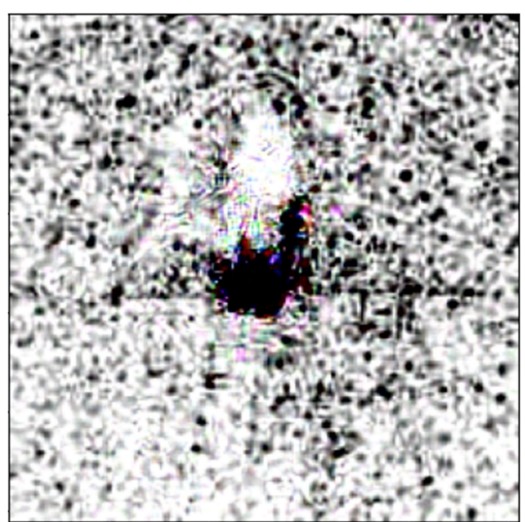

**Figure 7.** The pixel differences between the compressed and reconstructed images of the adversarial samples generated by the SFA method and the images generated solely using the SFA method are shown in color, with non-zero pixels highlighted for better visualization.

We refer to the SFA method with the added compression and reconstruction module as SFA-CR. In our comparative experiments, we used ResNet18 as a substitute model, VGG11 as the victim model, and evaluated the performance of the SFA and its variant methods against their corresponding -CR methods. The experimental results are shown in Figure 8.

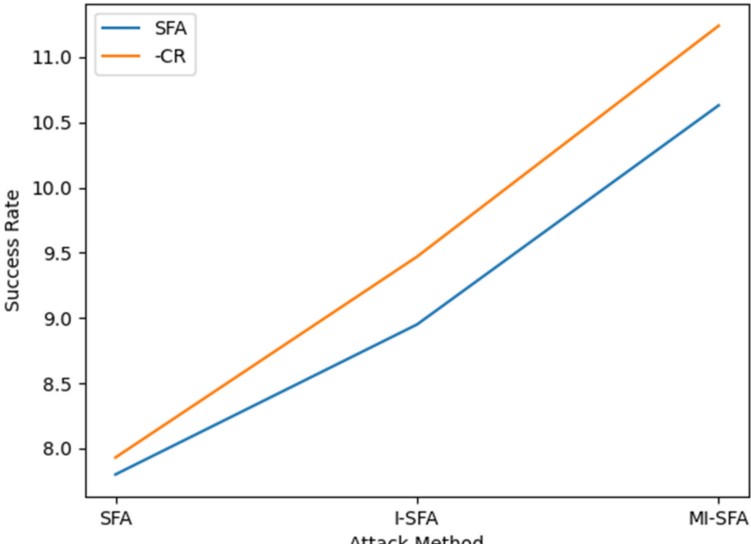

**Figure 8.** The comparison of attack success rates between SFA and its variant methods and their corresponding -CR methods is shown in the following line graph, where the yellow line represents the -CR methods and the blue line represents SFA and its variants.

Obviously, the adversarial effect is slightly enhanced when the adversarial images are compressed to PNG format and then reconstructed to ndarray. This enhancement is more significant when iterative attacks are used because the adversarial images are compressed and reconstructed after each iteration. Meanwhile, as shown in Figure 7, the changes made by compression and reconstruction mainly affect the adversarial regions with specific constraints, such as fitting shallow features in this paper. Therefore, the adversarial examples are relatively concentrated in the object regions of the image, and the changes made by compression and reconstruction are mainly reflected near the target contour. Thus, we speculate that compression and reconstruction methods may not be effective for ordinary human-eye-invisible adversarial perturbations without any constraints, but they will still be effective for adversarial patches.

## 6. Conclusions and Future Work

Overall, this paper designs a new adversarial attack method called SFA. Compared to traditional adversarial attack methods, this paper pays more attention to the impact of image changes in shallow features on the adversarial example. By introducing the critical features of the first pooling layer generated by the aggregated gradient, SFA proposes a new loss function that combines feature-level attack loss and end-to-end level attack loss. Experimental results show that compared with the corresponding baseline methods, our SFA method increases the success rate of targeted attack on a single model by an average of 3.73% under the black-box scenario and increases the success rate of ensemble model attack by an average of 4.61%. Through the SFA method proposed in this paper, the safety and defense of artificial intelligence in the SAR image field can be improved, and the discriminative ability of SAR recognition models can be enhanced, making the application of future deep neural networks on SAR images more robust.

In future work, we hope to explore and study the impact of compression and reconstruction transformations on adversarial examples at the feature level to identify the true reasons that affect the adversarial performance of adversarial examples. We also plan to study the possibility of reducing the number of ghost data packages while maintaining the attack effectiveness of SFA, in order to develop a more efficient SFA method.

**Author Contributions:** Conceptualization, G.L. and Z.P.; methodology, G.L.; software, G.L.; validation, G.L., D.Z. and X.Z.; formal analysis, G.L.; investigation, G.L., G.Z., T.L. and L.Z.; resources, Z.P.; data curation, G.L.; writing—original draft preparation, G.L.; writing—review and editing, W.B. and Y.D.; visualization, G.L.; supervision, Z.P. and Y.D.; project administration, Z.P.; funding acquisition, Z.P. All authors have read and agreed to the published version of the manuscript.

**Funding:** This research was supported by the National Natural Science Foundation of China, grant number 62076251.

**Data Availability Statement:** The data presented in this study are available on request from the corresponding author.

**Conflicts of Interest:** The authors declare no conflict of interest.

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
