# Peer review of "Boosting Adversarial Transferability with Shallow-Feature Attack on SAR Images"

_remotesensing, doi:10.3390/rs15102699_

Round 1

Reviewer 1 Report

1. In related works, the authors reviewed some attack methods including white box attack methods, black box attack methods and feature-level attack methods; but in Table 1, the authors only gave the examples of the first two types of attack methods. It would be better to give the example of feature-level attack methods.

2. In Table 1, some properties of the different methods are the same, it would be better to summary them than listing them separately. In other words, authors should find more difference of different attack methods.

3. In Table 3~7, some values are bolded and the corresponding bolded rule should be given. And the authors only listed these different results, the reasons of the experimental results in these tables should be interpreted instead of just listing their values.

4. In experiments, as shown in Table 2, twelve typical SAR-ATR models in Ref.[40-45] were employed, but these models were proposed before 2020. Although the recognition accuracy is over 96%, aren't there any better models that have been proposed in recent years?

There are some mistakes to be checked and modified.

Reviewer 3 Report

General comments:

The manuscript focuses on the image space and semantic information contained in the shallow features of SAR-ATR models, and constructs a feature-level loss function to enhance the transferability of generated adversarial examples. However, I think there are some major problems existed which need to be reconsidered and studied before resubmission.

Firstly, given that the manuscript 's title mentions "Boosting Adversarial Transferability on SAR Images", a comparison with recent adversarial attack methods proposed on SAR images would be more convincing, such as 1)’Adversarial deception against SAR target recognition network, IEEE JSTARS’; 2)’A Target-region-based SAR ATR Adversarial Deception Method, ICSIP 2022’.

Secondly, the success rate of the attacks in the entire experiment is generally low (less than 10%), and the success rate of the Baseline method is also generally low. Is this phenomenon specific to SAR images or is the success rate of transfer-based attacks generally low? Since the study focuses on transfer-based attacks, it would be interesting to compare the performance of some transfer-based attacks on SAR images.

Thirdly, since the attack target for all experiments in the manuscript is set to BTR60, it would be interesting to explore the effectiveness of the proposed method in attacking the other nine categories and whether its performance remains consistent.

Minor comments:

1. In the introduction of DNN applications, I suggest to replace the CV applications with some emerging Remote Sensing applications, such as ‘A Sidelobe-Aware Small Ship Detection Network for Synthetic Aperture Radar Imagery, IEEE TGRS’;’What Catch Your Attention in SAR Images: Saliency Detection Based on Soft-Superpixel Lacunarity Cue, IEEE TGRS’, etc..

2. Since the proposed methodology in this manuscript is characterized as a black-box targeted attack, the baseline methods used for comparison have demonstrated impressive performance in white-box untargeted attacks. However, it is important to investigate their performance in black-box targeted attacks and whether this would impact their success rate in transfer attacks.

3. Page 12 line 412, how were the parameters for the baseline method set, and are these settings optimal for each method?

4. Page 11 line 367,is the total loss function in the experiment simply the addition of the KL loss and CE loss ? Were the relative weights of the two losses adjusted using hyperparameters in the later experiments?

I recommend to further polish the English issue, i.e., reduce the usage of 'We'.

Reviewer 4 Report

This paper presents a black-box targeted attack method called Shallow-Feature-Attack (SFA) for adversarial example generation on Synthetic Aperture Radar (SAR) images. The authors propose to exploit shallow features of the model to generate adversarial examples, assuming that they are more capable of reflecting spatial and semantic information in the image. It is compared to several baseline methods (FGSM, I-FGSM, MI-FGSM, and Ensemble-Attack), and the experimental results demonstrate that the proposed method improves the success rate of single-model attack under black-box scenarios.

The paper is well-organized and well-written, making it easy to understand the proposed method and experimental setup. However, the paper does not provide a detailed analysis of the success of the SFA method compared to the baseline methods, explaining why the focus on shallow features contributes to the improved attack success rate (add more references for pg.2, line 73~76). Provide more insights or analysis on why the ensemble attack did not show significant improvement compared to single-model attacks. Also, add more reference for black box attack methods (pg. 4, line 169~173).

In general, white box attacks are based on complete knowledge of the target system, allowing for more targeted and effective exploitation, while black box attacks rely on observing the system's behavior without any knowledge of its inner workings, making them more representative of real-world attack scenarios but potentially less effective. Your paper is very interesting as it shows a higher success rate compared to white box attack-based models. However, it seems that you only compared deep learning-based models for surrogate and target systems. The Shallow Feature Attack (SFA) relatively simplifies deep learning models and reduces model complexity. Given this, I am wondering if you could also compare your approach with conventional machine learning-based models. This comparison may provide additional insights into the effectiveness of your proposed method across a wider range of models and could potentially strengthen your findings.

Overall, this paper presents an interesting method for generating targeted adversarial examples on SAR images. The experimental results demonstrate the effectiveness of the proposed method, and the paper is well-written and organized. However, the paper would benefit from a more in-depth analysis of the reasons for the success of the SFA method and additional experiments investigating the impact of compression and reconstruction transformations on adversarial examples.

pg.13 Table3 ~ 7, add the one more column for average (or sum) of success rate for each model.

pg.15 At the discussion, provide the time consumption for each model.

Round 2

Reviewer 1 Report

The manuscript has been revised and improved a lot, resulting in the proposal of being accepted.

  • There are still some spelling mistakes that need to be carefully checked and corrected.

Author Response

Dear Reviewer,

    I would like to express my gratitude once again for the valuable feedback you provided on our manuscript. With your help, our manuscript has undergone a significant transformation, and we are very grateful for your assistance. In addition, we have checked and revised the English descriptions in this article. Thank you for your feedback on our manuscript again.

Reviewer 2 Report

I suggest to accept it in present form.

Author Response

Dear Reviewer,

    I would like to express my gratitude once again for the valuable feedback you provided on our manuscript. With your help, our manuscript has undergone a significant transformation, and we are very grateful for your assistance.  Thank you for your feedback on our manuscript again.

Reviewer 3 Report

In General comments,

For response 1: clear,And it would be advantageous to compare the transferability of these adversarial attack methods in SAR Images and integrate your method into them.

For response 2: Black-box targeted attack is undoubtedly the most challenging aspect of adversarial attacks. The successful methods in optics are generally difficult to achieve the same effects in SAR. Maybe the success rate in this direction is generally at this level, and it is hoped that future work can further improve the attack success rate. It would be nice if the attack success rate can reach 40-50%.

For response 3:clear,it can be seen that your method can achieve a certain attack effect in other categories.

In Minor comments,

For response 1:clear.

For response 2:Just like the comparative experiment added to the SVA method later, your method goes even further in the FGSM-based method, which belongs to longitudinal comparison. If some transfer attacks from other factions in SAR domain can be added for horizontal comparison, the performance will be better.

For response 3:clear.

For response 4:clear.

This reviewer suggest to polish the English style.

Reviewer 4 Report

I am wrting to convey my satisfaction with the updated version of your manuscript, follwing my suggestions provided during the review process. It is evident that you have carefully considered and incorporated the recommendations, resulting in a stronger and more impactful piceo of work.  

I think that it is now well-prepared for publication 

Author Response

(The authors gave the same response as above.)
